# Procalcitonin to Guide Antibacterial Prescribing in Patients Hospitalised with COVID-19

**DOI:** 10.3390/antibiotics10091119

**Published:** 2021-09-17

**Authors:** Stephen Hughes, Nabeela Mughal, Luke S. P. Moore

**Affiliations:** 1Chelsea and Westminster Hospital NHS Foundation Trust, 369 Fulham Road, London SW10 9NH, UK; stephen.hughes10@nhs.net (S.H.); Nabeela.mughal@nhs.net (N.M.); 2North West London Pathology, Imperial College Healthcare NHS Trust, Fulham Palace Road, London W6 8RF, UK; 3NIHR Health Protection Research Unit in Healthcare Associated Infections and Antimicrobial Resistance, Imperial College London, Du Cane Road, London W12 0NN, UK

**Keywords:** procalcitonin, COVID-19, antimicrobial stewardship, bacterial co-infection

## Abstract

Antibacterial prescribing in patients presenting with COVID-19 remains discordant to rates of bacterial co-infection. Implementing diagnostic tests to exclude bacterial infection may aid reduction in antibacterial prescribing. (1) Method: A retrospective observational analysis was undertaken of all hospitalised patients with COVID-19 across a single-site NHS acute Trust (London, UK) from 1 December 2020 to 28 February 2021. Electronic patient records were used to identify patients, clinical data, and outcomes. Procalcitonin (PCT) serum assays, where available on admission, were analysed against electronic prescribing records for antibacterial prescribing to determine relationships with a negative PCT result (<25 mg/L) and antibacterial course length. (2) Results: Antibacterial agents were initiated on admission in 310/624 (49.7%) of patients presenting with COVID-19. A total of 33/74 (44.5%) patients with a negative PCT on admission had their treatment stopped within 24 h. A total of 6/49 (12.2%) patients were started on antibacterials, but a positive PCT saw their treatment stopped. Microbiologically confirmed bacterial infection was low (19/594; 3.2%) and no correlation was seen between PCT and culture positivity (*p* = 1). Lower mortality (15.6% vs. 31.4%; *p* = 0.049), length of hospital stay (7.9 days vs. 10.1 days; *p* = 0.044), and intensive care unit (ICU) admission (13.9% vs. 40.8%; *p* = 0.001) was noted among patients with low PCT. (3) Conclusions: This retrospective analysis of community acquired COVID-19 patients demonstrates the potential role of PCT in excluding bacterial co-infection. A negative PCT on admission correlates with shorter antimicrobial courses, early cessation of therapy, and predicts lower frequency of ICU admission. Low PCT may support decision making in cessation of antibacterials at the 48–72 h review.

## 1. Introduction

Early analysis of coronavirus disease (COVID-19) cohorts demonstrated a low incidence of microbiologically confirmed bacterial co-infection in patients presenting with community onset viral infection (3–5% in community onset infection, increasing among patients requiring more intensive healthcare interventions throughout their admission) [1,2,3]. Despite the data on low incidence of bacterial co-infection, high antibacterial prescribing is still evident in COVID-19 patient management [4].

Even with low rates of positive bacterial coinfection, without a method to distinguish at-risk patient groups, cautious over-prescribing of antibacterials will continue [5]. Work to date has highlighted the potential for traditional biomarkers (e.g., C-reactive protein (CRP) and neutrophils) to exclude bacterial co-infection but most of this work was completed before the introduction of dexamethasone in standard care [6].

Procalcitonin (PCT) is a 116-amino acid peptide, a precursor of calcitonin, and found in high concentrations in response to local inflammatory mediators, particularly due to bacterial endotoxins. It has been studied in patients with suspected or confirmed bacterial infection to help guide treatment response. PCT serum levels measured in systemic viral infections are markedly lower than in bacterial infections. It may, therefore, play an important role in differentiating bacterial from viral infective aetiology in patients presenting with unconfirmed infections, helping to reduce unnecessary antibacterial usage. Despite this, use of PCT in the UK is uncommon, and for routine management of bacterial infection, it is not thought to provide any additional benefit over more traditional biomarkers (e.g., CRP).

There has been a renewed interest in PCT during the COVID-19 pandemic, where patients are presenting with raised systemic inflammatory markers (e.g., CRP and white cell count (WCC)) independent of bacterial infection. Excluding bacterial infection in these acutely unwell patients, therefore, proves challenging. The use of PCT has been proposed as a more sensitive analysis of bacterial co-infection, but real-life data on PCT in COVID-19 is limited [7,8]. PCT may be elevated by non-bacterial causes including acute respiratory distress syndrome (ARDS) in COVID-19 patients; this limits the positive predictive value of PCT in this patient group. However, a negative PCT result in COVID-19 patients may offer antimicrobial stewardship (AMS) teams some utility in identifying patients with low probability of bacterial infection. This may enable early cessation of antibacterials at 48–72 h review, where a low probability of bacterial infection is present [9].

To analyse the impact of PCT as an AMS tool among patients presenting with COVID-19, we undertook a retrospective analysis of community onset COVID-19 managed in an acute hospital setting. We analysed the correlation with PCT and antibacterial prescribing as well as looking at the association with confirmed bacterial infections with serum PCT results. We aim to determine if PCT aids with the 48–72 h review of empiric therapy to minimise unnecessary antibacterial exposure. 

## 2. Materials and Methods

A retrospective observational analysis was undertaken of all hospitalised patients with COVID-19 across a single-site NHS acute Trust; Chelsea and Westminster Foundation Trust (London, UK). All patients with confirmed SARS-CoV-2 between 1 December 2020 and 28 February 2021 were included; patients where COVID-19 was suspected but not confirmed by respiratory tract PCT were excluded. Electronic patient records (Millenium^®^, Cerner Corp, Lees Summit, MO, USA, and ICNet^®^, Baxter, Gloucester, UK) and microbiology laboratory data (Sunquest ^®^ v8.3, AZ, USA) were used to identify patients, clinical data, and outcomes. Positive microbiological isolates (within 5 days of admission) that warranted targeted antibacterial treatment by the on-duty microbiology team were agreed to be a clinically important bacterial pathogen. Procalcitonin serum assays (Alinity i B·R·A·H·M·S, Abbott, VA, USA); values of 0.25 pg/mL at 6–24 h post-COVID diagnosis were defined as low risk of bacterial co-infection in community-onset COVID-19. Electronic prescribing records were analysed to identify antibacterial prescribing on admission and at 48–72 h review. The utility of a diagnostic test on antimicrobial stewardship (AMS) interventions in patients with early death (72 h from admission or SARS-CoV-2 detection) is unclear, therefore we have excluded this cohort from analysis of PCT usage.

All data were anonymised and collated using Excel 2017. Descriptive statistics were derived, using GraphPad^®^ (v8, 2018). Chi square and Fisher’s exact tests were used for analysis of categorical data and the Mann–Whitney U test for non-parametric continuous variables.

This project was registered as a service evaluation with the Chelsea & Westminster NHS Foundation Trust Antimicrobial Stewardship Committee (28 February 2021). Patient consent was waived, as this was completed as a retrospective analysis, completed as part of a service evaluation in line with local governance policy.

## 3. Results

A total of 730 patients with confirmed SARS-CoV-2 were identified during the study period; see Table 1. 

Antibacterials were initiated within 48 h of admission in 310/624 (49.7%) patients presenting with COVID-19 (Figure 1). On admission, 33/74 (44.5%) patients with a negative PCT on day 0/1 had their treatment stopped within 24 h. A total of 6/49 (12.2%) patients were started on antibacterials, but a positive PCT had their treatment stopped. In those patients continued on antibacterials beyond 72 h, a further 58/128 had PCT assays taken to guide empiric antibacterial therapy. Of these, 40/58 had PCT < 0.25 and 23/40 (57.5%) had their antimicrobials stopped within the subsequent 24 h.

Patients with PCT repeated on days 0 and 1 were analysed to assess the reliability of a single PCT assay result. A total of 5/16 patients with a reported negative PCT (<0.25 pg/mL) on admission had contradictory results (>0.25 pg/mL) in the proceeding 24 h (Figure 2). 

A low burden of community onset bacterial infection was evident (19/594; 3.2%),with significant bacterial isolates. High PCT values did not correlate with a likelihood of subsequent culture positive pyogenic infections, with 2/51 (3.9%) patients with a high PCT having a significant culture from blood/chest/urine, and 3/77 (3.9%) with a low PCT having a significant culture (*p* = 1; Table 2).

CRP correlates with procalcitonin in our cohort, with a 78 (46–119 IQR) and 170 (117–246) seen in patients with admission negative and positive PCT, respectively (*p* ≤ 0.0001). In 45 patients with admission (first 72 h) CRP < 50 and a follow-up admission PCT level, a negative PCT was identified in 37/45 (82.2%) of patients.

## 4. Discussion

This retrospective analysis of community-acquired COVID-19 patients admitted to a London hospital demonstrates the potential role of PCT in excluding concurrent bacterial co-infection. A negative PCT on admission correlates with shorter antimicrobial courses, early cessation of therapy, and predicts lower ITU admission during the admission. Initiation of empiric antibacterials is common in this and other external studies despite the confirmed low rates of bacterial co-infection in newly hospitalized COVID-19 patients. Yet, identifying the small cohort of patients that do benefit from antibacterials is challenging; therefore, wider usage of antibacterials occurs. The antimicrobial stewardship strategy must adapt to provide our busy clinicians with reliable diagnostic criteria for confirming or excluding bacterial infection in patients with COVID-19 infection. Within this study, we analyse the outcome of the 48–72 h review of empiric antibacterials when procalcitonin results are available for the clinical team. 

A low rate of confirmed bacterial co-infection was identified in this wave 2 cohort of hospitalised COVID-19 patients. Despite the widespread use of steroids, confirmed bacterial pathogens remain similar to our early wave 1 analysis [2]. Confirming true bacterial infection remains challenging in this cohort; the presence of a bacterial pathogen from many sample types (including sputum) may indicate true invasive infection or colonisation. Positive sterile samples (e.g., blood culture) are uncommon, even in invasive respiratory bacterial infections; therefore, the presence or exclusion of bacteraemia cannot be reliably used to exclude respiratory tract infections.

Difficulties identifying the small numbers of patients who may benefit from antibacterial results early in their presentation among the wider excess antibacterial usage remains a challenge. Procalcitonin has conflicting supporting evidence for its role in COVID-19 [7,10,11,12]. Early work from Heesom et al. during wave one of COVID-19 in Southampton (UK) focused on patients admitted to a critical care unit, and demonstrates lower overall antibacterial usage (2 days less) in patients with low PCT [7]. A PCT cut-off of <0.5 pg/mL was used in this study, as opposed to a lower cut-off of <0.25 pg/L in our practice. The Vanhomwegen et al. study also looked at PCT usage (cut-off of <0.5 pg/mL) in the critical care unit, but only investigated the association of PCT with confirmed bacterial infection [10]. The low yield of positive isolates in this small study was underpowered to demonstrate any association with PCT and bacterial co-infection; antibacterial usage was not studied. Pink et al. compared CRP with PCT to diagnose bacterial co-infection in patients with COVID-19 [11]. With higher rates of bacterial co-infection confirmed (32%), the receiver operating characteristic (ROC) curve for positive bacterial infection was similar for CRP and PCT (area under the curve of 0.86 and 0.88, respectively). PCT had a sensitivity of 91% and a specificity of 81% for the detection of secondary bacterial infection when a cut-off of 0.55 pg/mL was used (negative predictive value of 94%). Fabre et al. analysed 611 patients with COVID-19 analysed. PCT values in patients with proven, possible, and low-probability bacterial co-infection were analysed; an overlap in PCT values seen across the group highlighting that a positive PCT may be present due to non-bacterial causes in COVID-19 patients [12]. No association was demonstrated with antibacterial treatment duration in patients with possible or low-risk bacterial pneumonia, despite PCT usage. 

Concerns about raised PCT results in response to ARDs and/or severe COVID-19 infection have been suggested to limit its utility. Within our practice, we use PCT for its negative predictive value and the lower cut-off (0.25 pg/mL) was chosen to improve specificity in this primarily non-critical care population. Our data suggests a negative PCT is associated with better prognosis for the patient. Moreover, we find a temporal relationship with low PCT value, and subsequent early cessation of antibacterials. A high PCT appears to be an independent risk factor for poor patient outcomes, both in this study and previous work [13]. 

The reproducibility of PCT on admission was concerning. Early PCT sampling (<6 h of admission) may result in false negative results, with contradictory results seen at the 24 h follow up in a subset of our group. It is possible that some of the patients without follow-up PCT values may reflect falsely low results and detract from our projections. We continue to advise avoiding PCT assays on day 0 and advocate PCT sampling the day after admission, where the negative predictive value is expected to be more robust.

Our analysis contains numerous limitations. The lack of a definitive test to confirm or exclude bacterial co-infection does not allow us to calculate specificity and sensitivity for this test in cohort. The reliance on direct culture for confirming and excluding infection for respiratory based bacterial infections is known to be limited, and may underestimate the true incidence. Not all patients had a PCT value due to (a) limited reagent in early December 2020 and (b) guidelines only advised PCT testing when bacterial infection was suspected. Not all bacterial infections originate from the respiratory tract; therefore, patients on antibacterials for non-respiratory tract infection on admission have been excluded from the initial analysis. Follow up treatments of healthcare-associated infections are not discussed in detail, due to the heterogeneity of presentation; most hospital related bacterial infections are iatrogenic and related to ventilation or central line use.

This study does provide some supporting evidence for the use of PCT as part of the AMS strategy to limit unnecessary antibacterial usage. This retrospective study shares our real-life experiences with introducing a biomarker into practice, to enable clinicians to make more confident decisions related to antibacterial prescribing. It highlights some of the perils with reproducibility of PCT assays, and likely reflects the dynamic changes of PCT in response to bacterial infection. We advise avoidance of early PCT sampling as a result (<6 h of admission).

## 5. Conclusions

The ambition to reduce unnecessary antibacterials in patients presenting with COVID-19 infection continues. Identifying the patient groups that may benefit from antibacterial therapy is more challenging, but a PCT assay in the first 48 h may provide utility in excluding possible bacterial infection. Low PCT (<0.25 pg/mL) correlates with low probability of bacterial infection, and supports decision making to aid cessation of antibacterials at the 48–72 h review. The utility may be further increased as we see increasing usage of IL-6 inhibitors for COVID-19 management where CRP monitoring becomes obsolete.

## Figures and Tables

**Figure 1 antibiotics-10-01119-f001:**
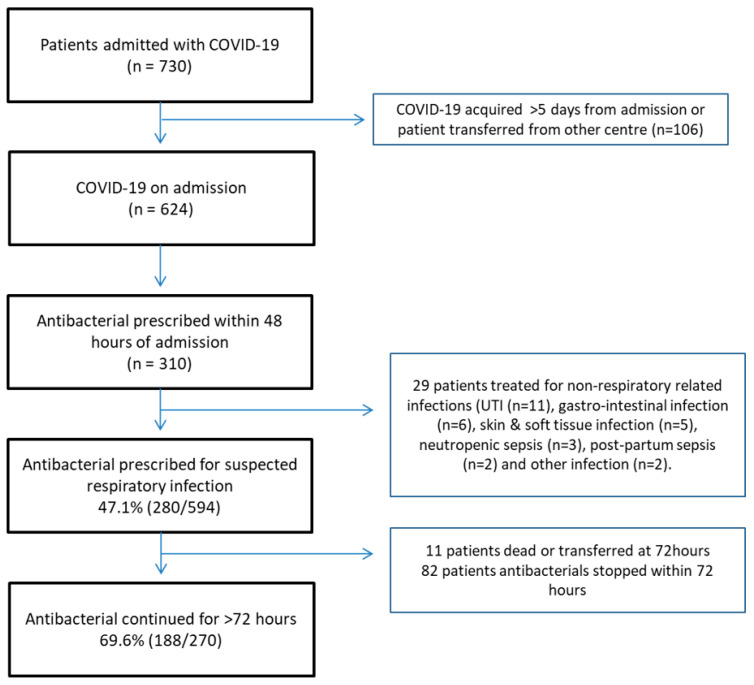
CONSORT diagram, associated of PCT and antibacterial prescription among COVID-19 patients, London, December 2020–February 2021 (UK wave 2).

**Figure 2 antibiotics-10-01119-f002:**
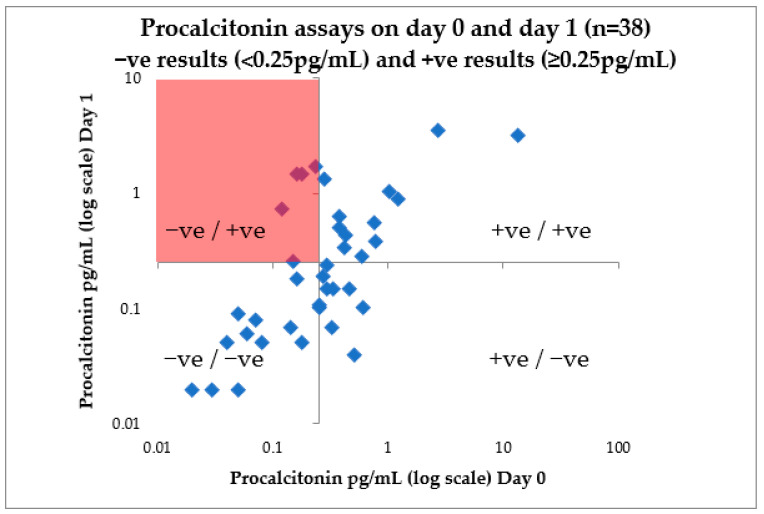
Reproducibility of procalcitonin taken on days 0 and day 1 of patient admission with COVID-19, London, December 2020–February 2021 (UK wave 2). Procalcitonin +ve results (defined locally as ≥0.25 pg/mL) and −ve results (defined locally as <0.25 pg/mL) are presented at day 0 (on admission) and day 1 (post-admission) to test reproducibility of this clinical test.

**Table 1 antibiotics-10-01119-t001:** COVID-19 patient and infection-related characteristics, London, December 2020–February 2021 (UK wave 2).

Subgroup	Antibacterials Initiated on Admission (n = 280)	Nil Antibacterials Initiated on Admission (n = 314)	*p*-Value
Age (IQR)	67.5yo (54.5–79.5)	62.2yp (40.2–78.5)	0.002
Sex (male/total)	151/280	153/314	0.218
Blood culture sent	200/280 (71%)	175/314 (56%)	<0.001
- +ve culture within 5 days of admission (of clinical importance)	1 *	0	-
- +ve culture after 5 days admission (of clinical importance)	13 #	3 $	
Respiratory MC&S sent	70/280 (25%)	41/314 (13%)	<0.001
- +ve culture within 5 days of admission (of clinical importance)	12 patients ##	5 patients $$	-
- +ve culture after 5 days admission (of clinical importance)	22 patients **	7 patients ~	
Legionella urinary antigen	59/280; all negative	32/314; all negative	0.004
Pneumococcal urinary antigen	52/280; 1 positive result	32/314; all negative	0.005
Viral Resp Screen	59/280; all negative	28/314; all negative	<0.001
Peak CRP in 72 h of admission (IQR)	83 (41–143)	63 (21–119)	<0.001
WCC on admission (IQR)	6.7 (5–9.6)	6.8 (4.8–9.3)	0.985
Treatment received during admission			
Steroids	225/280	15/314	<0.001
Remdesivir	99/280	6/314	<0.001
Favipirivir	33/280	4/314	<0.001
Tocilizumab	1/280	2/314	1
Initial ABX therapy		-	-
Amoxicillin +/− atypical antibacterial	204		
Co-amoxiclav +/− atypical antibacterial	10		
Ceftriaxone +/− atypical antibacterial	27		
Doxycycline monoRx	4		
Levofloxacin	26		
Piperacillin/tazobactam +/− atypical	6		
Other	3		
Systemic antifungals (any time during admission)	21 ***	1 $$$	-
In-hospital mortality at 30 days	54/280	63/314	0.179

* = *K. pneumoniae*; # = *E. faecalis* (×4), *E. faecium* (×2), *C. albican*, *C. glabrata*, *E.coli*, *H. alive*, *K. pneumonia*, MRSA (×2) & *Pseudomonas aeruginosa*; $ = *B. ovatus*, *E. faecium* × 2, *Candida* spp. × 3 & *S. marcesens*; ** = (*P. aeruginosa* × 8, *K. pneumonia* × 3, *C. koseri*, *K. aerogenes*, *H. alvei*, *E. faecium*, *M. morganii* (×2), MRSA × 4, *S. maltophilia* (×5), *S. marecscens*; ## = (*K. pneumonia* (×3), MSSA (×2), *S. pneumonia*, *P. aeruginosa* (×2), *P. mirabilis*, *M. morganii*, *E. cloacae*, *H. influenzae* ); $$ = (*E.coli*, MRSA, MSSA × 3, *H. influenzae* (mixed with MSSA); ~ (*Pseudomonas* spp. (×5), *K. aerogenes* and mixed (MRSA, *Raoltella* spp., *S. marcesens*, *S. maltophilia*). *** 2 × Voriconazole (proven Aspergillosis), 9 × Ambisome® (VAP empiric), 4 × Anidulafunign (suspected or proven invasive Candida infection), 6 × Fluconazole (suspected or proven invasive Candida infection); $$$ × Ambisome® (VAP empiric). Chi square and Fisher’s exact tests were used for analysis of categorical data and the Mann–Whitney U test for non-parametric continuous variables.

**Table 2 antibiotics-10-01119-t002:** Antibacterial usage on admission for patients presenting with community acquired pneumonia, London, December 2020–February 2021 (UK wave 2).

	PCT < 0.25 in 1st 48 h	PCT ≥ 0.25 in 1st 48 h	Significance
Initiated ABX on admission	77/127 (60.6%)	51/69 (73.9%)	*p* = 0.0835
ABX continued for >72 h	41/77 (53.2%)	44/51 (86.3%)	*p* ≤ 0.0001
Duration of ABX median (IQR) days	4 (1.5–7)	6 (4–9)	*p* = 0.00222
Peak CRP in first 72 h Median (IQR)	78 (46–119)	170 (117–246)	*p* ≤ 0.0001
WCC on admission Median (IQR)	6.1 (4.5–9.2)	7(4.65–10.35)	*p* = 0.34722
In-hospital mortality at 30 days	12/77 (15.6%)	16/51 (31.4%)	*p* = 0.0487
ITU admission (any time during admission)	11/79 (13.9%)	20/49 (40.8%)	*p* = 0.0012
Length of admission Median (IQR)	7.9 (4.85–14.9) days	10.1 (6.1–30.1) days	*p* = 0.04444
Any carbapenem usage during admission	6/127 (4.7%)	15/69 (21.7%)	*p* = 0.0005
Any systemic antifungal treatment during admission	5/127 (3.9%)	13/69 (18.8%)	*p* = 0.0012
CA-Bacteraemia	0	1 (*K. pneumoniae*)	-
Pneumococcal antigen positive	0	0	-
CA-culture positive bacteria in sputum	3 (*E. cloacae*, *M. morganii*, *P. mirabilis*)	1 (*P. aeruginosa*)	-
Follow up +ve PCT (After 5 days admission) *	4/25	14/33	*p* = 0.0452

IQR—inter quartile range; CRP—C reactive protein; WCC—white cell count; PCT—procalcitonin; CA = community acquired; ITU—intensive therapy unit. * A follow up PCT was defined as a PCT after 5 days admission that was greater than the admission PCT or represented a new PCT > 0.25 when admission testing was not completed. Chi square and Fisher’s exact tests were used for analysis of categorical data and Mann–Whitney U test for non-parametric continuous variables.

## Data Availability

The datasets used and/or analysed during the current study are available at https://doi.org/10.6084/m9.figshare.14838312.v1 (accessed on 17 August 2021).

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
