# Peer review of "Procalcitonin to Guide Antibacterial Prescribing in Patients Hospitalised with COVID-19"

_antibiotics, 2021, doi:10.3390/antibiotics10091119_

Round 1

Reviewer 1 Report

This manuscript addresses the use of procalcitonin as an indicator of bacterial infection in patients with COVID-19

There are somethings that should be taken in consideration:

  • In this section the authors should make a short explanation of the use of procalcitonin as indicator for bacterial infections.
  • Material and methods sections is misplaced as section 4, after the results that is section 2 and section 3 marked as discussion.
  • Information about information consent statement should be included in the material and methods section not as a separate statement.
  • Even though it is well known, the description of CRP should be specified the first time that it appears.
  • Results should be a little more explained in this section.

Author Response

Dear reviewer,

Thank you for your comments. Please find our responses below. 

Kind regards,

Stephen Hughes (on behalf of all the authors)

In this section the authors should make a short explanation of the use of procalcitonin as indicator for bacterial infections.

We have added some additional background information on PCT and its role in practice prior to the COVID19 pandemic in the introduction section

Material and methods sections is misplaced as section 4, after the results that is section 2 and section 3 marked as discussion.

We have moved this section to section 2

Information about information consent statement should be included in the material and methods section not as a separate statement.

We have updated this accordingly

Even though it is well known, the description of CRP should be specified the first time that it appears.

We have added the full description of CRP to the introduction section

Results should be a little more explained in this section.

Thank you - we have included a more detailed analysis of the results and comparison to other published data in our results

Reviewer 2 Report

In this study, authors explored the possible role of procalcitonin in prescribing antibacterials in COVID-19 patients.

This concise work provides some novel information in the field of extremely important topic - high antibacterial prescribing in COVID-19 patients. Manuscript is well written, however, before potential publication, some issues should be addressed:

In my opinion, this paper should be categorized as communication or short report, rather than original article, due to concise style of writing and provided results.

In Introduction section, more existing information regarding the potential role of procalcitonin should be provided (including possibly more information from ref. 8 and 9)

In the Results, p value should be limited to 3 decimal places

Table 1 is rather large – all unnecessary information, that are not important for this study should be removed.

In Table 2, abbreviation explanations are missing from footnotes

Line 105 – more explanations of “conflicting supporting evidence” should be provided

Author Response

Dear reviewer,

Thank you for your comments and suggestions. We agree with these and have made the changes accordingly.

Kind regards,

Stephen Hughes (on behalf of all the authors)

In my opinion, this paper should be categorized as communication or short report, rather than original article, due to concise style of writing and provided results.

We would be happy to present this as a short report.

In Introduction section, more existing information regarding the potential role of procalcitonin should be provided (including possibly more information from ref. 8 and 9)

We have added some additional background information on PCT and its role in practice prior to the COVID19 pandemic in the introduction section

In the Results, p value should be limited to 3 decimal places

We have changed all P values to 3 decimal places

Table 1 is rather large – all unnecessary information, that are not important for this study should be removed.

We have removed unnecessary data from the table and added it under the footnotes where relevant

In Table 2, abbreviation explanations are missing from footnotes

Thank you – we have update the abbreviations to this table

Line 105 – more explanations of “conflicting supporting evidence” should be provided

Thank you – we have provided a more detailed analysis of the supporting evidence for and against the routine use of PCT in this area.

Reviewer 3 Report

The paper deals with Procalcitonin to guide antibacterial prescribing in patients hospitalised with COVID-19. The idea is good, but the paper is superficially presented, and the form is extremely untidy; however I am convinced that the authors can complete it properly. In other words, both the form of presentation and the content of the paper are important in publishing it. Therefore, I present below my suggestions for improving the form and content of this paper but first please always check and respect the Instructions for authors https://www.mdpi.com/journal/antibiotics/instructions 

And when something is not clear about how to be written, just open an already published paper and all will be clear.

Form suggestions

Key words must be separated by semicolon, not by comma.

L143. Please use mL instead of ml (Litter is the international unit of measure for volume)

Content suggestions

I suggest as Keywords: procalcitonin; COVID-19; antimicrobial stewardship; bacterial co-infection.

Introduction section. It is very short. Background on the topic must be developed.

Please highlight as better is possible the aim of the study (as the last and separate paragraph of Introduction section). What makes special this study? Which is its novelty character or its special aspects? Why have the authors chosen this topic? What differentiate this paper from others in the same topic? Please make this aim of the study more relevant.

Table 1 (and Table 2)

  • Please complete the first cell of the table. It is not allowed empty cells in a scientific paper.
  • Last column: please insert p instead of Comment (in the head of the table, and let just the numerical values in all other cells of that column (in this way you avoid repetitions) (remove p = from the entire column)
  • Row containing Treatment received during admission: please merge all cells, in this way, there will be no empty cell on that row.
  • Remove empty row after Tocilizumab
  • the 2 cells above the last p value, if there is no value, please insert "-" symbol
  • centring some columns and choosing the text to be in the middle of the cells will improve the aspect of the table.
  • stay consistent on a certain type of annotation (use "p' in both Tables, not "P")

Both chapters of Results and Discussion are poorly developed and debated. You should analyse if procalcitonin has been recognized as a marker of covid bacterial co-infection in other studies and possibly make a table with the results obtained in those studies (in Discussion).  

What other types of medicine has been given to the patients in your study? Oral or intravenous corticosteroids? Was corticosteroid administration associated with an increase of procalcitonin because of bacterial infection in the context of immunosuppression? You can consult and refer to Behl T., et al. The dual impact of ACE2 in COVID-19 and ironical actions in geriatrics and paediatrics with possible therapeutic solutions. Life Sci. 2020, 257, 118075. https://doi.org/10.1016/j.lfs.2020.118075 ; Kabir M.T., et al. nCOVID-19 Pandemic from Molecular Pathogenesis to Potential Investigational Therapeutics, Front. Cell Dev. Biol. 2020, 8:616. https://doi.org/10.3389/fcell.2020.00616    

Furthermore, is the antiviral treatment associated with modification of PCT value? What type of antiviral treatment did the patients in your study received? How many of the patients received both antiviral and antibacterial? Did antiviral therapy reduce the risk of bacterial supra infection? You can check for ideas and refer to Negrut, N., et al. Efficiency of antiviral treatment in COVID 19. Exp. Ther. Med, 2021, 21, 648. https://doi.org/10.3892/etm.2021.10080

What were the isolated bacterial strains in patients with bacterial infection? What methods did you use to diagnose those infections? How did you know if it was or not a previous infection or an hospital acquired infection? Please see and refer to Zaha, D.C.; et al. Antibiotic Consumption and Microbiological Epidemiology in Surgery Departments: Results from a Single Study Center. Antibiotics 2020, 9, 81. https://doi.org/10.3390/antibiotics9020081 

Even the Conclusions section is not mandatory, I suggest being added as it allows in the best way to highlight the main findings of the research.

Author Response

Dear Reviewer

We thank you for your considered comments and feedback. We have taken this in to account and updated the manuscript accordingly. 

Regards

Key words must be separated by semicolon, not by comma.

Thank you – we have updated this accordingly

L143. Please use mL instead of ml (Litter is the international unit of measure for volume)

Thank you – we have updated this accordingly

I suggest as Keywords: procalcitonin; COVID-19; antimicrobial stewardship; bacterial co-infection.

Thank you – we have updated this accordingly

Please highlight as better is possible the aim of the study (as the last and separate paragraph of Introduction section). What makes special this study? Which is its novelty character or its special aspects? Why have the authors chosen this topic? What differentiate this paper from others in the same topic? Please make this aim of the study more relevant.

Thank you – we have added more explicit aims of this study and what this study adds to the available published literature

Introduction section. It is very short. Background on the topic must be developed.

We have developed the introduction / background to provide  the reader with more insight to the role of PCT in AMS and COVID-19

Table 1 (and Table 2)

Please complete the first cell of the table. It is not allowed empty cells in a scientific paper.

We have added content here

Last column: please insert p instead of Comment (in the head of the table, and let just the numerical values in all other cells of that column (in this way you avoid repetitions) (remove p = from the entire column)

Thank you – we have updated this accordingly

Row containing Treatment received during admission: please merge all cells, in this way, there will be no empty cell on that row.

Thank you – we have updated this accordingly

Remove empty row after Tocilizumab

Thank you – we have deleted this accordingly

the 2 cells above the last p value, if there is no value, please insert "-" symbol

Thank you – we have updated this accordingly

centring some columns and choosing the text to be in the middle of the cells will improve the aspect of the table.

Thank you – we have centred this accordingly

stay consistent on a certain type of annotation (use "p' in both Tables, not "P")

All p values are standardised in line with comment above

Both chapters of Results and Discussion are poorly developed and debated. You should analyse if procalcitonin has been recognized as a marker of covid bacterial co-infection in other studies and possibly make a table with the results obtained in those studies (in Discussion). 

Thank you for this – the rapidly emerging data around this important subject would warrant a more detailed analysis of all available studies. Here, we have now included a more discussion on the key papers published to date on PCT in COVID19 antibacterial use or bacterial co-infection. As this is not a comprehensive review of the available data, we have not tabulated this but rather made comparisons with our work

What other types of medicine has been given to the patients in your study? Oral or intravenous corticosteroids? Was corticosteroid administration associated with an increase of procalcitonin because of bacterial infection in the context of immunosuppression? You can consult and refer to Behl T., et al. The dual impact of ACE2 in COVID-19 and ironical actions in geriatrics and paediatrics with possible therapeutic solutions. Life Sci. 2020, 257, 118075. https://doi.org/10.1016/j.lfs.2020.118075 ; Kabir M.T., et al. nCOVID-19 Pandemic from Molecular Pathogenesis to Potential Investigational Therapeutics, Front. Cell Dev. Biol. 2020, 8:616. https://doi.org/10.3389/fcell.2020.00616  

The use of steroids for COVID 19 and other indications is detailed in table 1. We have not differentiated between intravenous and oral as many patients will have had interchangeable routes of administration and we do not think the route will impact on our primary outcome of this study. Similar dosing (6mg dexamethasone) is advised for both intravenous and oral steroid use in COVID19 therefore we would not anticipate changes in immunosuppressive burden.

The use of corticosteroids is likely a measure of disease severity and is expected to predict increase morbidity and mortality as a result. We know PCT results also predict poor clinical outcomes therefore some association is suspected but the authors do not think this is related to infective complications but rather a more severe presentation of COVID19

We have not investigated the use of other medications such as Angiotensin pathway inhibitors, anticoagulants or other non-evidence based therapies as whilst they may impact on COVID-19 survival but we did not expect to see any impact on bacterial co-infection rates. Antiviral and anti-IL-6 inhibitors were included to help readers relate this population to their own practice.

Furthermore, is the antiviral treatment associated with modification of PCT value? What type of antiviral treatment did the patients in your study received? How many of the patients received both antiviral and antibacterial? Did antiviral therapy reduce the risk of bacterial supra infection? You can check for ideas and refer to Negrut, N., et al. Efficiency of antiviral treatment in COVID 19. Exp. Ther. Med, 2021, 21, 648. https://doi.org/10.3892/etm.2021.10080

We have presented the antiviral usage within this cohort in table 1. We have details on the concurrent usage of antivirals and antibacterials for this group but have not presented any analysis on whether antiviral therapy if predictive of PCT or bacterial co-infection. We could not find any physiological rationale to link these two together on admission. We would predict patients that are responsive to antiviral therapy will have shorter hospital admission and lower need for critical care admission, reducing risk of secondary bacterial co-infection. This subject probably warrants a separate analysis in dedicated manuscript to address these issues and may offer a future line of research for this group but remains outside the scope for this manuscript.

What were the isolated bacterial strains in patients with bacterial infection? What methods did you use to diagnose those infections? How did you know if it was or not a previous infection or an hospital acquired infection? Please see and refer to Zaha, D.C.; et al. Antibiotic Consumption and Microbiological Epidemiology in Surgery Departments: Results from a Single Study Center. Antibiotics 20209, 81. https://doi.org/10.3390/antibiotics9020081 

The  methods and table 1 include how the bacterial pathogens were collated and presented. Definition of clinically important pathogens that warranted targeted antibacterial therapy has been added to the methodology.

Our definitions included bacterial pathogens within the first 5 days of admission only and were defined locally as community acquired pathogens. Previous infections may have been present (defined as recurring isolation of pathogenic bacteria) and were included

Even the Conclusions section is not mandatory, I suggest being added as it allows in the best way to highlight the main findings of the research.

We have inserted a summarising paragraph at the end of the discussion to highlight the major findings of the paper.

Round 2

Reviewer 2 Report

The authors have been responsive, and have responded adequatly to each of the comments. The work has been improved, and I don't have additional questions.

Author Response

We thank you for your time to review this and feedback

Reviewer 3 Report

2. Materials and Methods. Please detail the inclusion/exclusion criteria of the patients in your study. I understand that patients having confirmed SARS-CoV-2 were considered but other criteria? 

Table 1. Last cell in the table. Please remove p= (it is already mentioned in the head of the table that it is about p value that column).  

Between the text and references in the manuscript, an empty space must be inserted. Please check and revise the entire manuscript in this regard.

Figure 1 must be replaced with a better quality one. It is blurred. Same - the text on Figure 2. Additionally, please explain all the abbreviations used in the Figures 1 and 2 under the table (according to the Instructions for authors). Please also check that all abbreviations in the manuscript to be explained.

Some statements in the manuscript are not well sustained by references. Please check, revise and complete. Maybe some references of those I mentioned in my previous report (which authors totally ignored) can be useful. 

Having so many limitations (as even the authors state) which the authors describe at the final of Discussion section (L196-207), I suggest also that strengths of this study must be added and well highlighted in order to justify why to publish this paper.

References section. Please write the references in MDPI Style. In this regard, please check the Instructions for authors - Antibiotics journal. Please complete also all the information required for each reference.

6 self citation of LSP MOORE (ref. 1, 2, 5, 6, 11, 12) from a total of 15 references (meaning 40%) are too much. Please replace at least 4 of them (10% is usually allowed to be self citation in a professional paper as far as I know).

Author Response

Dear Reviewer,

Thank you for your comments and feedback. Please find our responses below.

Regards,

Stephen 

2. Materials and Methods. Please detail the inclusion/exclusion criteria of the patients in your study. I understand that patients having confirmed SARS-CoV-2 were considered but other criteria? 

We included all patients with a confirmed SARS-CoV-2 detection. Patients with suspected SARS-CoV2 but with no confirmed virology were excluded. We have updated the manuscript to highlight this

Table 1. Last cell in the table. Please remove p= (it is already mentioned in the head of the table that it is about p value that column).  

Thank you – we have actioned this change

Between the text and references in the manuscript, an empty space must be inserted. Please check and revise the entire manuscript in this regard.

Thank you – we have actioned this change

Figure 1 must be replaced with a better quality one. It is blurred. Same - the text on Figure 2. Additionally, please explain all the abbreviations used in the Figures 1 and 2 under the table (according to the Instructions for authors). Please also check that all abbreviations in the manuscript to be explained.

We have replaced the two figures with a higher definition photo. We have also provided more context for the abbreviations used in figure 2.

All abbreviations used now have an explanation with first use

Some statements in the manuscript are not well sustained by references. Please check, revise and complete. Maybe some references of those I mentioned in my previous report (which authors totally ignored) can be useful. 

We have added references to the new discussion points as requested.

We have looked at the reviewer’s four article suggests and found these subjects of interest. We had not used these texts in our write up and some were outside the scope of this manuscript.  As such we have not chosen to include these offered references for this paper

Having so many limitations (as even the authors state) which the authors describe at the final of Discussion section (L196-207), I suggest also that strengths of this study must be added and well highlighted in order to justify why to publish this paper.

Thank you – we will add this to the final discussion point (line 212)

References section. Please write the references in MDPI Style. In this regard, please check the Instructions for authors - Antibiotics journal. Please complete also all the information required for each reference.

Thank you – we have used the Mendeley reference tool for ‘Antibiotics’ journal to update our reference style

6 self citation of LSP MOORE (ref. 1, 2, 5, 6, 11, 12) from a total of 15 references (meaning 40%) are too much. Please replace at least 4 of them (10% is usually allowed to be self citation in a professional paper as far as I know).

We agree this appears somewhat excessive. The local research group had presented some of the early COVID19 AMS work and we have sought to reference this. We have removed references 6, 11 and 12 from the test as they do not add anything new.